# Basal Parasitic Fungi in Marine Food Webs—A Mystery Yet to Unravel

**DOI:** 10.3390/jof8020114

**Published:** 2022-01-26

**Authors:** Doris Ilicic, Hans-Peter Grossart

**Affiliations:** 1Leibniz Institute of Freshwater Ecology and Inland Fisheries, Alte Fischerhütte 2, 16775 Stechlin, Germany; doris.Ilicic@igb-berlin.de; 2Institute of Biochemistry and Biology, Potsdam University, Maulbeerallee 2, 14469 Potsdam, Germany

**Keywords:** basal fungi, parasites, Chytridiomycota, Rozellomycota, food web, biological carbon pump

## Abstract

Although aquatic and parasitic fungi have been well known for more than 100 years, they have only recently received increased awareness due to their key roles in microbial food webs and biogeochemical cycles. There is growing evidence indicating that fungi inhabit a wide range of marine habitats, from the deep sea all the way to surface waters, and recent advances in molecular tools, in particular metagenome approaches, reveal that their diversity is much greater and their ecological roles more important than previously considered. Parasitism constitutes one of the most widespread ecological interactions in nature, occurring in almost all environments. Despite that, the diversity of fungal parasites, their ecological functions, and, in particular their interactions with other microorganisms remain largely speculative, unexplored and are often missing from current theoretical concepts in marine ecology and biogeochemistry. In this review, we summarize and discuss recent research avenues on parasitic fungi and their ecological potential in marine ecosystems, e.g., the fungal shunt, and emphasize the need for further research.

## 1. Marine Fungi

Marine fungi are ubiquitous [1,2,3,4,5], occurring from Arctic and Antarctic [6,7,8] to tropical waters [9], inhabiting both surface waters and deep-sea sediments [10,11]. In addition, they exhibit a wide range of lifestyles, being saprotrophic, mutualistic or parasitic, and occur on a wide range of substrates [1,12,13,14,15]. To date, there have been various definitions as to what a “marine fungus” is. The term was first introduced by Barghoorn and Linder [16] who provided a foundation and a stimulus to the study of these organisms. Considering that many fungi detected in marine environments were already characterized from soil or plant habitats, Kohlmeyer and Kohlmeyer [17] have distinguished between obligate marine fungi which “grow and sporulate exclusively in a marine or estuarine habitat” and facultative marine fungi which “normally occupy terrestrial or freshwater habitats but are capable of growing and probably sporulating in a marine habitat” [18,19]. Jones et al. [20] took this a step further by including the term marine-derived fungi as “those taxa isolated [from marine habitats] during bioprospecting for new secondary metabolites”. In other words, all marine-derived fungi comprise those recovered from marine environments either through culturing or metagenomic methods, but whose obligate, or facultative marine nature, is not certain [13]. Finally, the most quoted and broad definition has been proposed by Pang et al. [2] who defined a marine fungus as “any fungus that is recovered repeatedly from marine habitats and: (1) is able to grow and/or sporulate (on substrata) in marine environments; (2) forms symbiotic relationships with other marine organisms; or (3) is shown to adapt and evolve at the genetic level and be metabolically active in marine environments”.

As with all fungi, marine fungi have traditionally been classified based on morphological features only [21,22]. However, such phenotypic classification is highly subjective, and does not say much on the evolutionary and ecological significance of the species in the kingdom Fungi. Johnson and Sparrow [23] classified fungi in oceans and estuaries into four classes: ‘Phycomycetes’, which included both fungi and fungi-like organisms [24]; ‘Fungi Imperfecti’, asexual morphs of the Ascomycota and the Basidiomycota [15]; and ‘Ascomycetes’ (Ascomycota) and ‘Basidiomycetes’ (Basidiomycota). Using phylogeny-based instead of traditional morphology-based classification has advanced our knowledge on the phylogenetic diversity of fungi [15,25]. As a result of genome data analysis [26,27], six phyla with marine representatives have been identified in the Kingdom Fungi: Ascomycota, Basidiomycota, Blastocladiomycota, Chytridiomycota, Glomeromycota and Mucoromycota [15]. In a more recent metabarcoding proteome analysis using whole-genome information, Rozellomycota (syn. Cryptomycota) was also found to be associated to the Kingdom Fungi [28]. The most recent overview of marine fungi taxonomy was done by Jones et al. [20] who summarized the classification of the known marine fungi (filamentous, zoosporic and yeasts) into Ascomycota, Basidiomycota, Blastocladiomycota, Chytridiomycota and Mucoromycota. Following often-used terminology, we can further organize these phyla into ‘lower fungi’ (also referred to as ‘basal fungi’) and ‘higher fungi’. The so-called lower marine fungi, reproduce by zoospores and include organisms of diverse affinity, i.e., members of the phyla Chytridiomycota [29], Blastocladiomycota [30] and Rozellomycota. Marine Ascomycota, Basidiomycota and Mucoromycota belong to higher marine fungi, which include most of the currently described species [17]. While the two groups differ in many important physiological and biochemical aspects, the important distinction is that the lower fungi reproduce asexually by releasing flagellated zoospores [31]. Zoospores are usually equipped with a single posteriorily-directed whiplash flagellum, although some species have multiflagellate zoospores [32] or, in at least one species of the Blastocladiales [33], the spores lack flagella [34]. Flagellated zoospores make lower fungi fully adapted to aquatic ecosystems by ensuring motility in the aquatic habitats [35,36]. In contrast, higher fungi reproduce via non-motile spores, such as ascospores or basidiospores that have evolved a fascinating array of morphological adaptations in order to survive in aquatic environments. One type of modification involves the presence of viscous, sticky appendages that evolved by fragmentation and/or extension of outer spore walls and that may enable spores to stick onto substrates in the moving water [37].

Microscopy- and culture-dependent approaches, such as growing organisms on prepared media or on incubated samples collected from the environment, have traditionally been used to study marine fungi [38]. Recent advances in DNA-sequencing technologies have led increasingly to the application of culture-independent methods, which involve direct isolation of the total microbial community DNA (the ‘metagenome’) and hence provide an opportunity to explore the “hidden” diversity of marine fungal communities [3,12,39]. Yet, such methods often yield terrestrial species which can reach marine ecosystems by air or can simply be washed away [40]. Consequently, although the shifting emphasis from cultivation-based studies to environmental DNA-based surveys has contributed significantly to our current understanding of marine fungal diversity and distributions, this shift has also led to unanticipated challenges, often hampering further progress [38]. To overcome the problems with interpreting results obtained using culture-independent methods, various other techniques are required to study such a diverse group of microorganisms, and this has led to a polarization of views when we pose the simple question “how many marine fungi are there?” At the moment, our knowledge of fungal diversity in marine ecosystems is still very limited. According to Hawksworth and Lücking [41], 2.2–3.8 million species of fungi inhabit the earth. This estimate takes into account rates and patterns at which new species are being described, evidence from environmental sequencing techniques, cryptic species and unexplored habitats. A total of 150,246 species of fungi have been described so far [42], out of which 1901 species of marine fungi are listed on the marine fungi website (https://www.marinefungi.org, accessed on 7 November 2021). Most have been discovered from metagenomes and lack any isolated strain. Hence, any species description and further distinction between obligate or facultative marine fungi is limited. Nevertheless, the increasing volume of research in recent years favors the idea that more new marine fungal taxa will be documented in the years to come [43]. The estimate of 10,000 as a total number of marine fungi proposed by Jones now seems more feasible, although we assume that the “real” number of marine fungi is much larger and includes a large variety of yet-undescribed species with unknown ecological function [9,44].

## 2. Parasitic Fungi in Marine Ecosystems

Parasitism is one of the earliest known and most common ecological interactions in nature [45], occurring in almost all environments. Nevertheless, parasites are often neglected in the management and conservation of biological resources and ecosystems [46]. Fungal parasites are small, short-lived, and rarely observed in the external environment [47], yet crucial for a comprehensive understanding of ecosystem dynamics. They have the potential to regulate host populations, mediate interspecific competition between hosts and other species, and affect community structure [39,43,45]. Despite their importance and ubiquitous nature, food web and ecosystem studies often ignore parasites. Yet, many recent studies suggest that parasites are important drivers of trophic interactions [47,48,49]; they affect the competitive ability for resource uptake among species in communities [50] and, since many parasites have free-living stages that can be grazed, they act not only as drivers of food chain properties but also as a trophic link, per se [51,52,53,54,55]. Moreover, parasites increase the resilience and persistence of ecosystems by promoting long food chains and multispecies connections known to stabilize the community [56]. In this review, the focus will be exclusively on marine fungal parasites.

Fungal parasitism is still a greatly neglected subject, although progress has been made during the last decades. Yet, the available studies are mainly related to freshwater ecosystems [43,45,48,49,50,51]. Environmental 18S rRNA surveys of microbial eukaryotes have recently revealed an unexpected diversity of major parasitic agents in aquatic systems—the so-called Dark Matter Fungi [57]—consisting primarily of members of the early diverging lineages within the fungal tree of life, such as Opisthosporidia [28,30,58] and Chytridiomycota [57,59]. Opisthosporidia is a super-phylum consisting of three main phyla: Aphelida, Rozellomycota and Microsporidia—also known as the ARM-clade [58]. Although some taxonomists place these intracellular parasites within Fungi [59,60,61], phylogenetic relationships are still mainly unresolved and considered to be an open question. While some studies indicate that Aphelida and Rozellomycota are a monophyletic group, placing them in a sister position to all other Fungi [58,62,63], other studies place these groups separately—Rozellomycota and Microsporidia at the basal position of Fungi, and Aphelida as sister of Blastocladiomycota [64]. Nevertheless, a unique characteristic of the ARM clade—the intracellular trophic stage which engulfs the host cell contents—strongly differentiates them from Chytridiomycota and other fungi [57]. Furthermore, parasitic fungi of marine phytoplankton were also found to belong to the Oomycetes [65]. Although they do not belong to the kingdom Fungi, they are referred to as “fungi-like” organisms and it is considered that they have similar ecological roles as true zoosporic fungi [21,65]. Parasites can also be found among the higher fungi but in this review [1,15,66] we focus only on the members of basal fungal phyla.

### 2.1. Aphelida

Aphelids comprise a small group of phagotrophic, intracellular parasites of algae [58], with four known genera: freshwater *Aphelidium* and *Amoebaphelidium*, and marine *Paraphelidium* and *Pseudaphelidium* (Table 1) [62]. Their morphology, life cycle, ecology, and taxonomy have been explored in several studies [58,62,67,68,69]. Their life cycle consists of a motile cell that is either flagellated (*Aphelidium*, *Pseudaphelidium*), amoeboid (*Amoebaphelidium*) or both (*Paraphelidium*) [67]. The infection begins when a zoospore attaches to the host algae and encyst. The encystment is followed by a germination and penetration of the host cell wall [58]. Inside the cyst a vacuole is formed. The vacuole enlarges and eventually pushes the content of the cyst into the host cell. At this stage of infection, the parasitoid transforms into a phagotrophic amoeba which engulfs the host cytoplasmic contents and gradually matures into a multinucleated plasmodium. When the host cytoplasm is consumed, the plasmodium divides into uninucleated zoospores which are then released through the penetration site [67]. In addition, some aphelid species have the ability to produce dormant or resting spores in order to survive the environmental extremes, and/or as the result of sexual reproduction [63]. Despite having just a few formally described species, Aphelida is highly diverse and includes many environmental sequences from diverse ecosystems [62].

### 2.2. Rozellomycota

Rozellomycota represents an understudied phylum of intracellular parasites of algae and water molds [70] that grow as naked protoplasts within their hosts [42]. Until now, only one genus (*Rozella*), consisting of less than 30 species, has been described (Table 1) [68]. *Rozella* was first introduced by Held [71] who briefly reported on morphological characteristics and ecology of some species known at that time [72]. Years later, Barr [73] placed these species according to the type of zoospores into the phylum Chytridiomycota. When James et al. [74] conducted a phylogenetic reassessment of zoosporic true fungi, *Rozella* was revealed to be a separate basal lineage to the core Chytridiomycota. Clustering of environmental sequences within the *Rozella* clade [60,75], followed by propositions of Jones et al. [60], the new phylum Cryptomycota (or, based on the only genus within, Rozellomycota) was introduced. Multiple studies have demonstrated their almost ubiquitous distribution in all habitat types [76,77]. In pelagic marine systems, Rozellomycota was reported in low but stable abundances [78,79]. Furthermore, their presence and significant abundances were reported also within benthic fungal communities [80,81]. Although environmental surveys have implied a rich biodiversity, it remains poorly characterized [60,82,83]. Microscopy-based studies have revealed a multistage life cycle consisting of an obligatory parasitic stage and a motile zoosporic stage [61,84]. The life cycle and morphology of Rozella resemble those of Aphelida; however, the two can be differentiated based on the host [58,68]. While the Aphelids infect algal hosts, Rozellomycota are mostly found as parasites of other zoosporic fungi and Oomycota [58]. According to Gleason et al. [72], the life cycle starts when a chemotactic zoospore attaches to the surface of the host cell and encysts. Encystment is followed by germination of infection along with penetration through a host’s cell wall. Inside the cyst, a vacuole swells, which pushes the naked amoeboid Rozella protoplast into the host cell. The protoplast grows in volume inside of the host cell and multiple nuclear divisions occur. Finally, the entire parasite cell will develop into a sporangium inside which zoospores are cleaved. The end of the cycle is indicated by the release of zoospores through one or several exit papillae.

### 2.3. Microsporidia

Microsporidia is a well-studied, diverse group of intracellular parasites, with a large number of fully sequenced genomes available [67]. Although it is considered that they infect only metazoans, some studies implicate that they present a much larger, yet still undescribed diversity [85]. Among documented marine Microsporidia, *Glugea* and *Pleistophora* are the most prevalent genera [86]. We present some other representatives in Table 1 and more information can be found in the literature [87,88]. These obligate parasites have a relatively uniform life cycle: a germinating spore injects the spore contents (sporoplasm) into the host cell via an injection tube. The sporoplasm grows into cells called meronts, which then further divide, producing the chitin cell wall, and develop into sporonts and then sporoblasts. Each sporoblast matures into a complex infective spore, representing a dispersal stage [58]. As a result of their lifestyle, the trophic stage contains very reduced genomes, lack of motile structures and true mitochondria, which are reduced to organelles called mitosomes [67]. Since mitosomes are unable to produce ATP, Microsporidia have developed a unique capacity to get ATP directly from the host cell via an array of horizontally acquired genes and, as such, they became “energy parasites” [58]. Due to their simplified cellular morphology and lack of mitochondria, Microsporidia were first thought to be early-branching eukaryotes that diverged before the acquisition of mitochondria. Their phylogenetic relatedness to fungi lays in the description of mitosomes and several mitochondria-related genes in their genomes. Today, Microsporidia are officially adopted by mycologists, although the nomenclatural rules for this group follow those of protists instead of the classical botanical rules that apply to other fungi [67].

### 2.4. Chytridiomycota

Chytridiomycota is another basal group within the kingdom Fungi and the members are presumed to retain key characteristics of the last common ancestor of Fungi and Animals [26,89]. These include a unicellular body bounded by a cell wall, which matures into a sporangium, within which many posteriorly-uniflagellate zoospores develop [45,90]. Thus, these fungi, commonly known as chytrids, thrive and reproduce in freshwater and marine environments, but can also be found commonly in habitats such as terrestrial soils where they need only a periodic film of surface water suitable for dissemination of zoospores [91]. It is known that chytrids reproduce asexually and the life cycle begins when zoospore attaches itself to a host cell wall by producing rhizoids [54]. As the cell feeds on the host, growth of the cytoplasm and numerous mitotic divisions will occur [32,35]. The zoospore encysts by forming a cell wall around the original membrane which then grows as the protoplasm increases in volume [29,32]. The enlarged cell becomes the zoosporangium and cleavage of the protoplasm: in particular, fusion of vesicles produced by a Golgi apparatus. This results in the production of individual zoospores that are released through a predetermined pore [92]. Flagellated zoospores swim for a short period of time in search of a suitable substrate. If such a substrate is located, they retract the flagellum, secret a cell wall to encyst and the cycle starts again [35,51,93]. If they do not find a suitable substrate in time, they die.

To date, an increasing number of studies are reporting the presence of chytrids in the marine environment [21,80,84,94,95,96,97,98], but a limited number have been properly identified and characterized [21]. These are species of *Rhizophydium* [99], *Thalassochytrium* [100] and *Chytridium* [101], which are either facultative or obligate parasites of macro-algae and invertebrates but diatoms (Table 1) [102,103]. All four genera have been placed in the phylum Chytridiomycota but the phylogenetic relationships of these ecotypes to other members of this phylum are not fully understood due to the lack of molecular data [104]. Following the initial work by Canter and Lund [105,106] and some later studies [107,108,109,110], chytrids are raising renewed interest, as further evidence accumulates for their widespread distribution across climatic regions, in both marine and freshwater ecosystems [36].

### 2.5. Oomycota—Fungi-like Organisms

Many species of heterotrophic stramenopiles such as representatives of the Oomycota (oomycetes) are common in the marine environment and known to infect several marine macroalgal species and planktonic diatoms [21]. In particular, oomycetes infecting marine phytoplankton comprise several marine representatives such as *Lagenisma coscinodisci* Drebes, which was reported as an endobiotic parasite of the centric diatom *Coscinodiscus centralis* Ehrenberg from the North Sea [111]. Furthermore, the endoparasitic, saprolegniaceous oomycete *Ectrogella* Zopf is a parasite in diatoms and, according to Sparrow [112], outbreaks of *Ectrogella perforans* Petersen may attain epidemic proportions in the marine pennate diatom *Licmophora* Agardh [113]. Other representatives can be found in Table 1. Oomycota, as other zoosporic fungi, reproduce asexually by means of flagellated zoospores that are produced in zoosporangia [21,113,114], and, as members of the ARM clade, they have been recognized as endoparasites [115]. Despite similarities in life cycles, it is important to highlight the fundamental difference between oomycetes and zoosporic fungi—oomycetes are stramenopiles, a heterotrophic sister group of, e.g., brown algae and diatoms [104]. Hence, oomycetes have predominantly cellulosic walls [113] in contrast to zoosporic fungi that are characterized by chitinaceous cell walls. Although oomycetes have frequently been reported in marine environments [7,21], not much is known about their biology and ecology. However, these parasitoids are known to play a significant role in breaking down blooms of their hosts, i.e., diatoms, and might also play similar roles in the marine food web as those of zoosporic true fungi [115].

## 3. Host Specificity

When discussing fungal parasitism, we refer to chytrids (Chytridiomycota), since they have been used as a model system in most research that has been undertaken until now on this topic.

The degree to which chytrids are specific to certain host species, genera or families remains an unresolved question. It has been suggested that chytrids are often highly host-specific [80,137,138], but researchers have also shown that within a single chytrid species both specialist and generalist strains coexist [48,139]. Nevertheless, our current knowledge is biased by the fact that both molecular and morphological identification are not sufficient or accurate in identifying chytrids at the species level [36]. What indicates the specific attachments of zoospores onto a particular host group is their chemotactic behavior [19,29,119]. Muehlstein et al. [140] showed that marine chytrid species (*Rhizophydium*) are positively attracted to amino acids and carbohydrates, both being photosynthetic by-products. That being said, light arguably has an important effect on the outcome of chytrid infection [29,119]. By affecting the photosynthetic rates of the host, light indirectly impacts chemotactic attraction of chytrid zoospores, which is itself driven by exudation of saccharides released as products of photosynthesis [141]. Furthermore, it is known that cell exudates are also responsible for chemotaxis and host recognition by chytrid zoospores [119]. Yoneya et al. [142] suggested a possible role of volatile organic compounds (VOC), released from microalgae and/or attached bacteria, as the infochemicals that allow chytrid zoospores to distinguish between host and non-host phytoplankton species.

## 4. Eco-Evolutionary Dynamics

It is important to highlight that any host-parasite relationship is a dynamic one and can evolve rapidly. Although it is not within the scope of this review to discuss the entire fields of chytrid biology, ecology and evolution, we will briefly address the importance of eco-evolutionary dynamics of fungal parasites and their hosts. Driven by fluctuating selection in host-parasite systems, oscillations of genotype abundances occur in time and every genotype can temporarily be best adapted [143]. This phenomenon is known as Red Queen Hypothesis, indicating that parasites have an important effect on (host) community composition dynamics in a given ecosystem. Such rapidly evolving host-parasite interactions also strongly depend on prevailing environmental conditions and as such, functionally link evolutionary and ecological processes. Further information can be found in the literature [109,143,144,145,146].

## 5. Diversity in Marine Ecosystems

Numbers of studies addressing diversity and abundances of Chytridiomycota are increasing with the improvement of molecular tools, in particular metagenomics approaches. Yet, information on their ecological role remains scarce. Efforts to retrieve such information due to the limited availability of model systems in culture [39,93,147] have been mainly focused on freshwater systems with a few, more recent, marine studies. Studies using next-generation sequencing technologies have revealed the prevalence of Chytridiomycota in both arctic and temperate marine habitats. Masana et al. [148] reported that Chytridimycota accounted for more than 60% of the rDNA sequences sampled in six near-shore sites in Europe, and were the most abundant fungal group in Arctic and sub-Arctic coastal habitats [6,80,89]. Although there are large numbers of described species of parasitic chytrids [20], only a few parasitic chytrid species have been genome sequenced and their phylogenetic positions clarified: *Rhizophydium littoreum* [120], *Thalassochytrium gracilariopsis* and *Chytridium polysiphoniae* (=*Algochytrops polysiphoniae*) [15,104]. Gaps in the reference databases relating to taxonomic coverage and marker coverage and the general lack of high quality, long sequence data are some of the major constraints for Chytridiomycota taxonomy [149,150]. Since fungi are phylogenetically diverse, DNA metabarcoding studies typically use markers that vary depending on the taxonomic group of interest and the resolution desired [149]. To overcome these drawbacks, Heeger et al. [150] created a long-read (ca. 4500 bp) bioinformatics pipeline that results in rates of sequencing error and chimera detection that are comparable to typical short-read analyses. The approach thus enabled the use of three different rRNA gene reference databases, thereby providing significant improvements in taxonomic classification over any single gene marker approach. Nevertheless, it is difficult to determine species composition and function (e.g., parasitic or saprotrophic) by analyzing environmental DNA alone [15]. Culturing, single cell PCR methods and whole genome sequencing [39,151] will presumably improve the representation of chytrids in future sequence databases.

## 6. Ecological Roles of Fungal Parasites

Besides their ubiquitous nature, the importance of parasites lies also in the ecological roles they hold. By regulating host populations and mediating interspecific competition between hosts and other species [152], they affect community structure [153], but can also alter biochemical cycles, change productivity, increase trophic chain length and number of links [52], and cause changes in the topology of the trophic network and functioning of the ecosystem [154]. Thus, fungal parasites, infecting phytoplankton as primary producers, change the flow of carbon (C) in aquatic ecosystems. This process, named “mycoloop” [155], proposes that fungal infections of phytoplankton hosts transfer once inaccessible organic matter (OM) from large, inedible hosts to zooplankton by producing zoospores [51,55,154]. Zoospores are rich in polyunsaturated fatty acids (PUFAs) and cholesterol [34,45,50,52,55,156], they have relatively low carbon to nutrient ratios [54,157] and synthesize sterols de novo [53], which makes them a nutrient rich food source for grazers such as zooplankton. Hence, the presence of chytrids may not only affect the quantity of food that is being transferred, but also its quality [54,55]. It has been suggested that fungal infections may also modulate algal-bacterial interactions [158]. By utilizing their phytoplankton hosts and causing massive cell lysis, fungi provide C substrates [110] in terms of dissolved organic matter (DOM) and therefore activate the microbial loop [38,159]. In contrast, Klawonn et al. [160] have recently demonstrated that this may not be the case since fungal parasites very efficiently utilize their hosts’ cellular contents and that C and N compounds are most efficiently transferred to attached sporangia, and the developed zoospores therein. Consequently, the overall photosynthetically active biomass is reduced, as is the phytoplankton-derived contribution to the dissolved organic carbon (DOC) pool. This process, where photosynthetic carbon is bypassed from the microbial loop to fungal parasites has been called “fungal shunt” and it promotes zooplankton-mediated over microbe-mediated remineralization [160]. Multiple lines of evidence suggest that fungi play important ecological roles in marine ecosystems, yet our knowledge is still scarce. Current research is mostly focused on freshwater habitats, and we can only speculate how these results apply for the fungal parasites inhabiting marine waters.

## 7. Biological Carbon Pump

The biological carbon pump comprises phytoplankton cells, their consumers and the bacteria that assimilate their waste, which play a central role in the global carbon cycle by delivering carbon from the atmosphere to the deep sea, where it is concentrated and sequestered for centuries [161,162]. Photosynthetically active phytoplankton in the euphotic zone (0–200 m depth) transforms dissolved inorganic carbon (DIC) to organic carbon, both dissolved and particulate forms [162]. The dissolved fraction is mainly respired by bacteria and the remaining refractory matter (RDOM) sinks and is mixed into the deep sea via the so-called microbial carbon pump (MCP) [163]. Considering the high efficiency of fungal parasites in utilizing OM from their hosts through the fungal shunt [160], infections can lead to substantially lower the contribution of phytoplankton derived OM to the DOM pool, which would consequently decrease the efficiency of the microbial carbon pump. Nevertheless, the particulate fraction is more significant when it comes to carbon export to the deeper layers of the ocean [139]. It is known that phytoplankton releases transparent polymeric particles (TEP)—gel-like, sticky particles predominantly composed of acidic polysaccharides [164]. Due to their high abundance in seawater and their surface reactivity, TEP scavenge trace elements (e.g., Fe and Th) and are the key agents for increasing the coagulation efficiency of physical aggregation processes [165]. Thus, TEP have an important role in biogeochemical fluxes and consequently, the efficiency of the biological carbon pump [166]. Mass flocculation and subsequent sedimentation of phytoplankton, especially diatoms, as large, rapidly sinking aggregates occur ocean-wide and represent a major global sink for carbon. It has been shown that these larger, recalcitrant phytoplankton cells serve as preferential hosts for fungal parasites such as chytrids [26,44,141]. Grossart et al. [39] introduced the term “mycoflux” which refers to any fungal interaction leading to aggregation or disintegration of organic matter. On the one hand, by taking up the nutrients from their hosts, chytrids may decrease the exudation of TEP and thus, indirectly affect aggregation processes leading to a decreased efficiency of carbon sequestration. On the other hand, fungal infections are often lethal and lead to fragmentation of large phytoplankton cells [36,45,167], making them more edible to zooplankton which then substantially contribute to carbon pump efficiency by excretion of fast-sinking fecal pellets. The overall transfer efficiency of the biological carbon pump is therefore determined by a combination of different factors: seasonality; composition of phytoplankton species; fragmentation of particles by zooplankton; solubilization of particles by microbes [168]; and presumably presence of fungal parasites on phytoplankton species [39,157] (Figure 1).

As shown in this review, fungal parasites significantly change the fate of photosynthetically derived carbon. Infecting large phytoplankton cells, they potentially circumvent the microbial carbon pump and consequently direct the carbon from phytoplankton to zooplankton, either via (1) mycoloop, mycoflux, and fungal shunt, or by (2) active fragmentation of their hosts. Thus, it is important to include fungal parasites into general biogeochemical concepts, e.g., the biological carbon pump and C and N cycling, since they significantly affect the efficiency of carbon sequestration, i.e., controlling the magnitude of the overall organic matter sinking flux. Our understanding of ocean biogeochemistry is of pivotal importance for climate and human-induced changes in food web dynamics, biogeochemical cycles and their feedbacks to future climate. Thus, incorporating fungal parasites into concepts and models is necessary for developing a more integrated view of the ocean carbon cycle, in particular the biological carbon pump efficiency, to better predicting and thus mitigating the negative effects of current global change.

## 8. Summary Points

Severe reference database gaps and need for long sequence data are still a major limitation when studying fungal parasites;Fungal parasites are important components of marine food webs;Transfer efficiency of the biological carbon pump is highly affected by the presence of fungal parasites on phytoplankton cells (both directions);We are missing enough model systems to study the ecology of parasitic fungi;Linkage of different OMICS tools, i.e., metagenomics, transcriptomics, proteomics and metabolomics could give us better insights into their ecological importance.

## 9. Future Perspectives

Further tool development by linking OMICS tools and experiments;Rapid changes in environmental factors due to global warming can have major effects on fungal parasites in marine ecosystems, either indirectly, by affecting their phytoplankton hosts, or directly, by affecting the parasites themselves;Thus, we highlight the need for more global and systematic ecological studies of fungal parasites in marine ecosystems.

## Figures and Tables

**Figure 1 jof-08-00114-f001:**
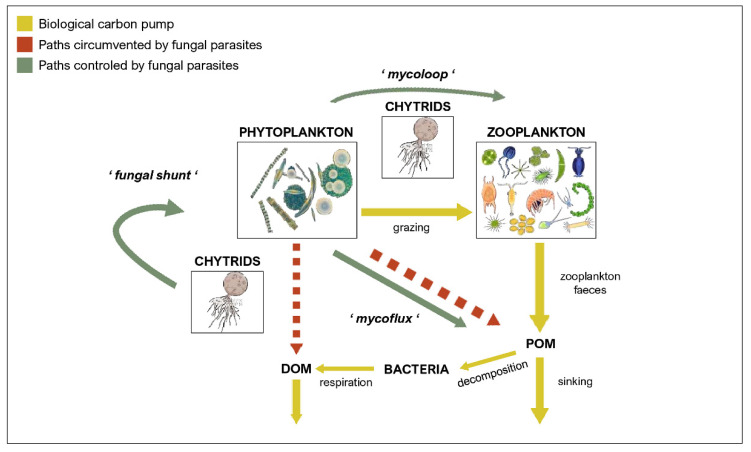
Fungal parasites are key components of the biological carbon pump. Fungal parasites take up phytoplankton-derived photosynthetic carbon (‘fungal shunt’) and thus lower the contribution to the DOM and POM pools. Through ‘mycoflux’, fungi control POM aggregation process, either decreasing (parasitic fungi), or increasing (saprotrophic fungi) the aggregation rate by promoting active aggregation via hyphae growth. Via fragmentation of large phytoplankton cells and redirecting carbon directly to zooplankton (‘mycoloop’) they can also indirectly modulate carbon sequestration via sinking zooplankton faeces.

**Table 1 jof-08-00114-t001:** Overview of marine parasites belonging to basal fungal phyla and their hosts.

	Parasite	Host	Literature
Aphelida	*Pseudaphelidium drebesii*	*Thalassiosira punctigera*	[116]
Chytrids	*Chytridium megastomum*	*Ceramium*	[117]
	*Chytridium polysiphonae*	*Sphacelaria, Pyaiella, Centroceras*	[66,104,117]
	*Coenomyces sp*	*Cladophora*	[66]
	*Dinomyces arenysensis*	*Dinoflagellates*	[99]
	*Olpidium rostiferum*	*Cladophora, Pseudo-nitzschia,*	[66,104,118]
	*Rhizophydium*	*Nitzschia, Rhizosolenia, Chaetoceros*	[119]
	*Rhizophydium aestuarii*	*Codium fragile*	[99]
	*Rhizophydium globosum*	*-*	[120]
	*Rhizophydium littoreum*	*Codium, Cancer anthonyi, Bryopsis*	[99,117,121]
	*Thalassochytrium gracilariopsis*	*Gracilariopsis*	[99,100]
Microsporidia	*Loma trichiuri*	*Trichiurus savala*	[88]
	*Microsporidium aplysiae*	*Aplysia californica*	[122]
	*Microsporidium cerebralis*	*Salmo salar*	[88]
	*Nematocenator marisprofundi*	*Desmodora marci*	[123]
	*Nosema pariacanthi*	*Priacanthus boops*	[88]
	*Oogranate pervascens*	*Maculaura sp.*	[88]
	*Pleistophora finisterrensis*	*Micromesistius poutassou*	[124]
	*Pleistophora hippoglossoideos*	*Hippoglossoides limandoides*	[88]
	*Pleistophora littoralis*	*Blennius pholis*	[125]
	*Pleistophora senegalensis*	*Sparus aurata*	[126]
	*Sporanauta perivermis*	*Odontophora rectangula*	[87]
	*Thelohania butleri*	*Pandalus jordani*	[127]
	*Unikaryon legeri*	*Meiogymnophallus minutus*	[128]
Oomyocta	*Cryothecomonas longipes*	*Thalassiosira, Pirsonia, Rhizosolenia*	[129]
	*Diatomophthora drebesii = Olpidiopsis drebesii*	*Rhizosolenia imbricata*	[115,130]
	*Ectrogella eurychasmoides*	*Licmophora lyngbyei*	[131]
	*Ectrogella marina*	*Chlorodendron subsalsum*	[103]
	*Ectrogella perforans*	*Fragilaria, Licmophora, Podocystis, Striatella, Synedra, Thalassionema*	[132]
	*Eurychasma dicksonii*	*Ectocarpus, Feldmannia, Punctaria, Pylaiella, Stictyosiphon, Striaria*	[23]
	*Eurychasmidium tumefaciens*	*Ceramium*	[117]
	*Lagenisma coscinodisci*	*Coscinodiscus*	[111,133,134]
	*Miracula helgolandica*	*Pseudo-nitzschia pungens*	[130]
	*Olpidiopsis porphyrae*	*Bangia, Porphyra*	[117]
	*Petersenia lobata*	*Aglaothamnion, Callithamnion, Ceramium, Gymnothamnion, Herposiphonia, Polysiphonia, Pylaiella, Seirospora, Spermothamnion*	[117]
	*Petersenia palmariae*	*Palmaria mollis*	[135]
	*Petersenia pollagaster*	*Chondrus*	[117]
	*Pontisma antithamnionis*	*Antithamnion*	[117]
	*Pontisma feldmannii*	*Falkenbergia, Trailliella*	[117]
	*Pontisma lagenidioides*	*Ceramium, Chaetomorpha, Valoniopsis*	[66]
	*Pythium marinum*	*-*	[117]
	*Pythium porphyrae*	*Porphyra*	[117]
	*Sirolpidium andreei*	*Acrosiphonia, Ceramium, Ectocarpus, Spongomorpha*	[117]
	*Sirolpidium bryopsidis*	*Cladophora, Rhizoclonium*	[117]
Rozellomycota	*Rozella marina*	*Chytridium polysiphoniae*	[136]

## Data Availability

Not applicable.

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
