# Peer review of "Basal Parasitic Fungi in Marine Food Webs—A Mystery Yet to Unravel"

_jof, 2022, doi:10.3390/jof8020114_

Round 1
Reviewer 1 Report
This paper focuses on the topic of fungal parasites, particularly on the basal fungal clades e.g. chytrids, Rozellomycota, which have importance in carbon cycling in marine ecosystems. This is an interesting paper that will attract a wide audience interested in marine fungi due to its large information how the parasitic basal fungi to partake in the biological carbon pump through these processes including mycoloop, mycoflux and fungal shunt. As we known, some members of higher fungi, i.e. Ascomycota and Basidiomycota, are also parasitic to marine algae or animals and play important roles in the ocean (Li et al., 2010, Acta Oceanol Sin, doi:10.1007/s13131-010-0065-4). Obviously, this issue is not a component of this paper. Thus, ‘parasitic basal fungi’ instead of ‘parasitic fungi’ in the title will be better in line with the content of this paper. Beside above, other concerns of mine can be found as the follows.
Line 62: don’t agree this sentence, marine fungi are not taxonomical groups, Jones [19] just summarized marine fungi according to taxonomical systems.
Line 67: replace ‘phylum’ with ‘phyla’.
Line 115-116: please delete this sentence, although I like this metaphor.
Line 120-122: please delete this sentence.
Line 167-168: please add reference.
Line 188-189: please concretize the difference on hosts.
Line 244: most marine green algae are macro-algae, so suggest delete ‘marine green algae’.
Reviewer 2 Report
Reviewers Comments:
The paper jof-1564912 ‘‘Parasitic fungi in marine food webs – a mystery yet to unravel’’. The results are attractive to the scientist relevant to this area ad overall authors have presented the results nicely.
However, the writing requires some modification.
Please add some references from recent literature.
Add a table and list common patristic fungi in marine ecosystems.
Add tables for Aphelida, Rozellomycota, Microsporidia, Chytridiomycota, and Oomycota – fungi-like organisms
What is the significance of the biological carbon pump and its relevance to this article? Please explain.
Figure 1: Is this figure made by authors? If not then add source.
